# Mode Localization and Eigenfrequency Curve Veerings of Two Overhanged Beams

**DOI:** 10.3390/mi12030324

**Published:** 2021-03-19

**Authors:** Yin Zhang, Yuri Petrov, Ya-pu Zhao

**Affiliations:** 1State Key Laboratory of Nonlinear Mechanics (LNM), Institute of Mechanics, Chinese Academy of Sciences, Beijing 100190, China; yzhao@imech.ac.cn; 2School of Engineering Science, University of Chinese Academy of Sciences, Beijing 100049, China; 3Faculty of Mathematics and Mechanics, St-Petersburg State University, 198504 St-Petersburg, Russia; yp@YP1004.spb.edu; 4Institute of Problems of Mechanical Engineering, 199178 St-Petersburg, Russia

**Keywords:** overhang, mode localization, eigenfrequency curve veering, mode splitting

## Abstract

Overhang provides a simple but effective way of coupling (sub)structures, which has been widely adopted in the applications of optomechanics, electromechanics, mass sensing resonators, etc. Despite its simplicity, an overhanging structure demonstrates rich and complex dynamics such as mode splitting, localization and eigenfrequency veering. When an eigenfrequency veering occurs, two eigenfrequencies are very close to each other, and the error associated with the numerical discretization procedure can lead to wrong and unphysical computational results. A method of computing the eigenfrequency of two overhanging beams, which involves no numerical discretization procedure, is analytically derived. Based on the method, the mode localization and eigenfrequency veering of the overhanging beams are systematically studied and their variation patterns are summarized. The effects of the overhang geometry and beam mechanical properties on the eigenfrequency veering are also identified.

## 1. Introduction

The model of the diatomic chain with perfect regularity, which assumes a simple cubic lattice with two different masses and the nearest neighbor interaction only, predicts that its frequency spectrum consists of two parts: The upper optical band and the lower acoustic band [1,2]. The effect of the introduction of an impurity atom into a perfect crystal is to bring a local defect, which changes the mass and stiffness at that given point. The existence of a local defect generates two new eigenfrequencies, which are isolated from both the optical and acoustic bands [1,2]. The normal modes associated with these two new eigenfrequencies are the localized modes [1], in which vibration concentrates around the local defect. The mechanism for the localized modes can be roughly given as follows: When a crystal vibrates with these isolated eigenfrequencies, which are the forbidden bands for most of the atoms, those atoms keep motionless except that of the local defect and few neighboring ones vibrating with much smaller amplitudes [1]. As early as in 1877, Lord Rayleigh’s study on the effect of the parameter variations on dynamical systems showed that under right circumstances a vibration mode can split away from the band of modes and exists above the maximum frequency of the unperturbed system [3]. In optics, the localized mode was firstly observed by Schaefer [4] in his experiment on the absorption spectrum of potassium chloride (KCl) at the frequencies above the maximum phonon frequency, in which a sharp localized optic mode appears due to the presence of a hydrogen ion impurity. In the study of electron transport, Anderson’s model [5] (nowadays known as the Anderson localization) shows that the electron motions becomes localized once an impurity is introduced into an “unperturbed” system, which can lead to the transition of a metallic conductor to a semiconductor or an insulator. In acoustics, the localized modes are realized by the so-called locally resonant sonic materials, which consist of two different materials: hard lead particle coated with soft silicone rubber [6]. The transmission coefficient of the lead–rubber composite structure has the sudden decreases at two different frequencies of an incidence wave, which are referred to as the dip frequencies [6]. At the lower dip frequency, the lead particle is similar to a mass in an oscillator, which moves as a whole as a rigid body; at the same time, the rubber displacements except those around the interface are very small. At the higher dip frequency, things are reversed: the lead particle now has almost no motion and most of the vibration is in the rubber. The motions at these two dip frequencies are referred to as the localized modes [6]. In these localized modes, most vibrations are confined either in the rubber layer or as the oscillation of the lead particle [6]; almost no waves are transmitted into the lead particle, i.e. the particle has no deformation. This localized mode provides another effective way of sound attenuation rather than absorption [6]. In optomechanics, a strong optomechanical coupling results in a localized mode, in which the reflection/absorption of photons or microwave phonons is dramatically changed [7]. Similar to in solid mechanics, the mode localization is also encountered once the periodicity/cyclicity of a structure is broken by a defect/irregularity [8,9].

Mode localization and veering are closely related [10]. Some scholars think that they are the two manifestations of a same drastic phenomenon [11]. Veering describes the phenomenon that two eigenvalues, which can physically correspond to either eigenfrequencies [10,11] or buckling loads [12,13], rapidly approach each other and then diverge abruptly without crossing. To emphasize this non-crossing property, veering is also variously called avoided crossing [14] or anti-crossing [15]. Coupling is the mechanism responsible for the mode localization and veering [16]. When two eigenfrequencies are well-separated, the effect of mode coupling is usually modest, and the modal motions are essentially independent from each other [17]. The mode coupling can be enhanced when the ratio of two eigenfrequencies is an integer, which is also called harmonic [18]. However, most structures with small linear vibrations do not have the harmonic property; various methods, such as the step-like design [19], are used to make a cantilever beam harmonic. Mode coupling can also be enhanced in nonlinear regime, in which large motion induced tension can effectively change the system stiffness and, therefore, tune the system eigenfrequencies to be harmonic [17,20]. Another alternative approach to enhance the mode coupling is to bring the two eigenfrequencies closer because veering indicates strong mode coupling [15]. For example, two modes of a square membrane with the eigenfrequency difference of 0.07% have been used to achieve a strong mode coupling in a cavity optomechanical system [21]. In the Kuramoto model of coupled oscillators, the effective strength of coupling is the paramount parameter of determining the synchronization [22] and pattern formation [23] in a dynamical system. However, extracting the effective strength of coupling directly from a real physical system is rather difficult. An advantage of studying the eigenfrequency curve veering is to provide an efficient way of evaluating/obtaining the effective strength of coupling.

The coupling between the optical and mechanical modes can result in either the damping or anti-damping force for either mode [24]. The damping effect has been used to cool down the thermal motion of a macroscopic resonator to the quantum ground state [25]. Quality factor is an essential factor determining the mass sensitivity in a mass resonator sensor application [26,27,28]. The quality factor of a mechanical resonator can be significantly improved by the anti-damping due to the optomechanical coupling [29]; the cavity optomechanical system is a possible candidate for the ultimate limit of mass detection [7,30]. One of the fast developing trends in optomechanics is to replace the optical cavity with an electrical one, which leads to electromechanics. When the materials of the mechanical end mirror are highly transparent to light, the radiation pressure exerted by photons can be weak, which is the means of the optomechanical coupling [31]. For example, graphene is 98% transparent to light, but it is basically opaque to the electric field [31]. Therefore, instead of an optical cavity, a microwave-regime electric cavity is introduced [7,31]. The vibration of the graphene resonator forms a movable capacitance together with a substrate; the electrical and mechanical energies are thus coupled [31]. Another trend is to replace the optical or electrical cavity by a higher mode of the mechanical resonator [32,33]. Overhang is a simple but effective structure to realize a strong coupling between modes by bringing the two neighboring eigenfrequencies closer. For example, Okamoto et al. [32] realized a strong mode coupling by using two modes of overhanging beams with the two closely spaced eigenfrequencies of 293.94 and 294.37 kHz, which also achieves a highly efficient energy transfer between the two modes. As shown in Figure 1, overhang is a mechanical sledge shared by two or several (sub)structures [34,35,36]. Overhang physically couples the (sub)structures and leads to the eigenfrequency veering and mode localization, which has been used as an effective mechanism to increase the sensitivity of mass sensing [35,36,37,38]. Furthermore, a veering locus also (approximately) corresponds to the maximum shift of an eigenfrequency [37], which is a much sought-after property in a mass resonator sensor [26,27,28].

Due to their simple design, overhanging structures are an important type of resonator structure [32,33,34,35,36,37,38,39,40,41]. Various numerical methods have been used to study the mode localization and eigenfrequency veering of the overhanging structures. However, most numerical discretization methods create or destroy the self-adjointness property of a continuous system [42], which causes error and even unphysical (un)coupling, and therefore, leads to wrongful results. This problem becomes much more serious when two eigenfrequencies are very close to each other in the veering region [43]. Even for a numerical discretization method which preserves the (non)self-adjointness property, such as the Galerkin method [42], wrongful computational results can still be obtained, as demonstrated by Leissa [44]. The finite element method fails in the computation of the three overhanging cantilevers due to the discretization asymmetry, which physically is induced by an adsorbate [41]. The asymptotic series and perturbation method can also break down as they assume slight changes in the modes [43]. Once a numerical discretization method is used for the computation of eigenfrequency veering, its validity is always a heated debate [42,44,45]. In this study, the matrix for the eigenfrequency computation is analytically derived and the computation is carried out without any numerical discretization. A comprehensive study on the variations of the eigenfrequencies and mode shapes of two overhanging clamped–clamped beams is presented. In contrast with the deceptive simplicity of the two overhanging beams, rich variation patterns of the mode localization, eigenfrequencies and their veerings are revealed. For each eigenfrequency of a uniform beam, a new and larger eigenfrequency is generated due to the coupling effect of an overhanging structure, which resembles the above case of lattice dynamics [1,2]. As the overhang length increases, this newly generated eigenfrequency increases much more quickly, which causes the separation of the two eigenfrequencies. Consequently, the modes associated with these two separating eigenfrequencies undergo the so-called mode splitting [14]. Furthermore, this new eigenfrequency also approaches the eigenfrequency of a higher mode and then diverges, which results in veering. The effects of the overhang geometry on the veering and mode localization are systematically studied. This comprehensive study should be of some help to the design of the overhanging resonator structures [32,33,34,35,36,37,38,39,40,41]. Recently, Kasai et al. [46] proposed an efficient method called “virtual coupling”, which can effectively change the coupling strength of a system through a nonlinear feedback scheme. Their method has great potential to be applied to various coupled systems. Besides designing the overhanging structures, this paper also presents a rather general study on the properties of coupled systems, which can provide some help to other coupled systems, such as optomechanical [21,24,29,30,31] and electromechanical [33,47,48] ones.

## 2. Model Development

In the schematic diagram of Figure 1, two beams are connected by two overhangs, which are clamped. $L_{1}$, *b* and *h* and $L_{2}$, $b^{*}$ and $h^{*}$ are the lengths, widths and thicknesses of the two overhangs, respectively. The widths and thicknesses of the two beams are $b_{1}$ and $h_{1}$ and $b_{2}$ and $h_{2}$, respectively. The kinetic energy (*T*) of the system is given as follows
(1)$$T=\int_{0}^{L_{1}}\frac{m}{2}{\left(\frac{\partial w}{\partial t}\right)}^{2}dx+\int_{L_{1}}^{L-L_{2}}\left[\frac{m_{1}}{2}{\left(\frac{\partial w_{1}}{\partial t}\right)}^{2}+\frac{m_{2}}{2}{\left(\frac{\partial w_{2}}{\partial t}\right)}^{2}\right]dx+\int_{L-L_{2}}^{L}\frac{m^{*}}{2}{\left(\frac{\partial w^{*}}{\partial t}\right)}^{2}dx,$$
where *L* is the length of the whole structure; *m* and $m^{*}$ are the mass per unit length of the two overhangs; and $m_{1}$ and $m_{2}$ are the mass per unit length of Beams 1 and 2, respectively. For the rectangular sections, m=ρbt, $m^{*}=\rho^{*}b^{*}t^{*}$, $m_{1}=\rho_{1}b_{1}t_{1}$ and $m_{2}=\rho_{2}b_{2}t_{2}$ (ρ and $\rho^{*}$ and $\rho_{1}$ and $\rho_{2}$ are the densities of the two overhangs and two beams, respectively). The first and third terms of Equation (Equation 1) are the kinetic energies of the two overhangs; the second term is those of the two beams. The corresponding transverse displacements of overhangs and beams are *w* and $w^{*}$ and $w_{1}$ and $w_{2}$, respectively.

The potential/bending energy (*U*) of the system is as follows: (2)$$U=\int_{0}^{L_{1}}\frac{EI}{2}{\left(\frac{\partial^{2}w}{\partial x^{2}}\right)}^{2}dx+\int_{L_{1}}^{L-L_{2}}\left[\frac{E_{1}I_{1}}{2}{\left(\frac{\partial^{2}w_{1}}{\partial x^{2}}\right)}^{2}+\frac{E_{2}I_{2}}{2}{\left(\frac{\partial^{2}w_{2}}{\partial x^{2}}\right)}^{2}\right]dx+\int_{L-L_{1}}^{L}\frac{E^{*}I^{*}}{2}{\left(\frac{\partial^{2}w^{*}}{\partial x^{2}}\right)}^{2}dx,$$
where *E* and $E^{*}$ and $E_{1}$ and $E_{2}$ are the Young’s moduli of the overhangs and beams, respectively. *I* and $I^{*}$ and $I_{1}$ and $I_{2}$ are the second moments of area of the overhangs and beams, respectively. For rectangular cross sections, $I=bh^{3}/12$, $I^{*}=b^{*}h^{*3}/12$, $I_{1}=b_{1}h_{1}^{3}/12$ and $I_{2}=b_{2}h_{2}^{3}/12$. By applying Hamilton’s principle, i.e., $\delta \int_{t_{1}}^{t_{2}}\left(T-U\right)dt=0$, the following governing equations are derived:(3)$$\begin{array}{c} \left\{\begin{array}{c} m\frac{\partial^{2}w}{\partial t^{2}}+EI\frac{\partial^{4}w}{\partial x^{4}}\;=\;0,\;0\leq x\leq L_{1}\\ m_{1}\frac{\partial^{2}w_{1}}{\partial t^{2}}+E_{1}I_{1}\frac{\partial^{4}w_{1}}{\partial x^{4}}\;=\;0,\;L_{1}\leq x\leq L-L_{2}\\ m_{2}\frac{\partial^{2}w_{2}}{\partial t^{2}}+E_{2}I_{2}\frac{\partial^{4}w_{2}}{\partial x^{4}}\;=\;0,\;L_{1}\leq x\leq L-L_{2}\\ m^{*}\frac{\partial^{2}w^{*}}{\partial t^{2}}+E^{*}I^{*}\frac{\partial^{4}w^{*}}{\partial x^{4}}\;=\;0,\;L-L_{2}\leq x\leq L. \end{array}\right. \end{array}$$

The following sixteen boundary conditions are also derived from Hamilton’s principle: (4)$$\begin{array}{c} w\left(0\right)=0,\;\frac{\partial w}{\partial x}\left(0\right)=0,\;w^{*}\left(L\right)=0,\;\frac{\partial w^{*}}{\partial x}\left(L\right)=0,\\ w\left(L_{1}\right)=w_{1}\left(L_{1}\right),\;\frac{\partial w}{\partial x}\left(L_{1}\right)=\frac{\partial w_{1}}{\partial x}\left(L_{1}\right),\;w\left(L_{1}\right)=w_{2}\left(L_{1}\right),\;\frac{\partial w}{\partial x}\left(L_{1}\right)=\frac{\partial w_{2}}{\partial x}\left(L_{1}\right),\\ EI\frac{\partial^{2}w}{\partial x^{2}}\left(L_{1}\right)=E_{1}I_{1}\frac{\partial^{2}w_{1}}{\partial x^{2}}\left(L_{1}\right)+E_{2}I_{2}\frac{\partial^{2}w_{2}}{\partial x^{2}}\left(L_{1}\right),\;EI\frac{\partial^{3}w}{\partial x^{3}}\left(L_{1}\right)=E_{1}I_{1}\frac{\partial^{3}w_{1}}{\partial x^{3}}\left(L_{1}\right)+E_{2}I_{2}\frac{\partial^{3}w_{2}}{\partial x^{3}}\left(L_{1}\right),\\ w_{1}\left(L-L_{2}\right)=w^{*}\left(L-L_{2}\right),\;\frac{\partial w_{1}}{\partial x}\left(L-L_{2}\right)=\frac{\partial w^{*}}{\partial x}\left(L-L_{2}\right),\\ w_{2}\left(L-L_{2}\right)=w^{*}\left(L-L_{2}\right),\;\frac{\partial w_{2}}{\partial x}\left(L-L_{2}\right)=\frac{\partial w^{*}}{\partial x}\left(L-L_{2}\right),\\ E_{1}I_{1}\frac{\partial^{2}w_{1}}{\partial x^{2}}\left(L-L_{2}\right)+E_{2}I_{2}\frac{\partial^{2}w_{2}}{\partial x^{2}}\left(L-L_{2}\right)=E^{*}I^{*}\frac{\partial^{2}w^{*}}{\partial x^{2}}\left(L-L_{2}\right),\\ E_{1}I_{1}\frac{\partial^{3}w_{1}}{\partial x^{3}}\left(L-L_{2}\right)+E_{2}I_{2}\frac{\partial^{3}w_{2}}{\partial x^{3}}\left(L-L_{2}\right)=E^{*}I^{*}\frac{\partial^{3}w^{*}}{\partial x^{3}}\left(L-L_{2}\right). \end{array}$$

The first four equations are the clamped boundary conditions for the two overhangs. The other twelve are at the two connecting points at $L_{1}$ and $L-L_{2}$. Physically, these twelve equations are to ensure the continuity of displacement, slope, moment and shear force at the two connecting points [19]. The governing equation of the overhang–beams–overhang structure is divided into four in three domains, as indicated by Equation (Equation 3). In the domain of $L_{1}\leq x\leq L-L_{2}$, the two beams can have two different vibrations of $w_{1}$ and $w_{2}$. However, the two overhangs are the parts shared by the two beams and, therefore, for each overhang, there is only one vibration, i.e., *w* for Overhang 1 and $w^{*}$ for Overhang 2. The two overhangs couple the two beams as they share the same displacement in the overhang sections. This coupling is also reflected in the boundary conditions at the two connecting points, as given in Equation (Equation 4). It is also noteworthy that Equations (Equation 1)–(Equation 4) are derived by a rather general method, which can be directly used to derive the governing equations of other type of overhanging structure with a small modification [49].

The following dimensionless quantities are introduced [50]
(5)$$\xi \;=\;\frac{x}{L},\;\xi_{1}\;=\;\frac{L_{1}}{L},\;\xi_{2}\;=\;\frac{L_{2}}{L},\;W\;=\;\frac{w}{L},\;W^{*}\;=\;\frac{w^{*}}{L},\;W_{1}\;=\;\frac{w_{1}}{L},\;W_{2}\;=\;\frac{w_{2}}{L},\;\tau \;=\;\sqrt{\frac{E_{1}I_{1}}{m_{1}L^{4}}}t.$$

The governing equations of Equation (Equation 3) are now non-dimensionalized as the following:(6)$$\begin{array}{c} \left\{\begin{array}{c} \alpha \frac{\partial^{2}W}{\partial \tau^{2}}+\gamma \frac{\partial^{4}W}{\partial \xi^{4}}\;=\;0,\;0\leq \xi \leq \xi_{1}\\ \frac{\partial^{2}W_{1}}{\partial \tau^{2}}+\frac{\partial^{4}W_{1}}{\partial \xi^{4}}\;=\;0,\;\xi_{1}\leq \xi \leq 1-\xi_{2}\\ \left(1+\Delta_{1}\right)\frac{\partial^{2}W_{2}}{\partial \tau^{2}}+\left(1+\Delta_{2}\right)\frac{\partial^{4}W_{2}}{\partial \xi^{4}}\;=\;0,\;\xi_{1}\leq \xi \leq 1-\xi_{2}\\ \alpha^{*}\frac{\partial^{2}W^{*}}{\partial \tau^{2}}+\gamma^{*}\frac{\partial^{4}W^{*}}{\partial \xi^{4}}\;=\;0,\;1-\xi_{2}\leq \xi \leq 1. \end{array}\right. \end{array}$$

The dimensionless quantities in Equation (Equation 6) are defined as follows:(7)$$\begin{array}{c} \;\alpha =\frac{m}{m_{1}},\;\gamma =\frac{EI}{E_{1}I_{1}},\;\alpha^{*}=\frac{m^{*}}{m_{1}},\;\gamma^{*}=\frac{E^{*}I^{*}}{E_{1}I_{1}},\;\Delta_{1}=\frac{m_{2}}{m_{1}}-1,\;\Delta_{2}=\frac{E_{2}I_{2}}{E_{1}I_{1}}-1. \end{array}$$

Physically, α and γ are the dimensionless mass per unit length and bending stiffness of Overhang 1 as compared with those of Beam 1. $\alpha^{*}$ and $\gamma^{*}$ are those of Overhang 2. $\Delta_{1}$ and $\Delta_{2}$ are the Beam 2 (dimensionless) deviations of mass per unit length and bending stiffness from those of Beam 1, respectively. By assuming $W=U\left(\xi \right)e^{i\omega \tau }$, $W^{*}=U^{*}\left(\xi \right)e^{i\omega \tau }$, $W_{1}=U_{1}\left(\xi \right)e^{i\omega \tau }$ and $W_{2}=U_{2}\left(\xi \right)e^{i\omega \tau }$ (ω ia the dimensionless circular frequency) and substituting them into Equation (Equation 6), the following solution forms are obtained
(8)$$\begin{array}{c} \left\{\begin{array}{c} U\;=\;A_{1}\sin\left(\kappa \beta \xi \right)+A_{2}\cos\left(\kappa \beta \xi \right)+A_{3}\sinh\left(\kappa \beta \xi \right)+A_{4}\cosh\left(\kappa \beta \xi \right),\;0\leq \xi \leq \xi_{1}\\ U_{1}\;=\;B_{1}\sin\left(\beta \xi \right)+B_{2}\cos\left(\beta \xi \right)+B_{3}\sinh\left(\beta \xi \right)+B_{4}\cosh\left(\beta \xi \right),\;\xi_{1}\leq \xi \leq 1-\xi_{2}\\ U_{2}\;=\;C_{1}\sin\left(\kappa_{2}\beta \xi \right)+C_{2}\cos\left(\kappa_{2}\beta \xi \right)+C_{3}\sinh\left(\kappa_{2}\beta \xi \right)+C_{4}\cosh\left(\kappa_{2}\beta \xi \right),\;\xi_{1}\leq \xi \leq 1-\xi_{2}\\ U^{*}\;=\;D_{1}\sin\left(\kappa^{*}\beta \xi \right)+D_{2}\cos\left(\kappa^{*}\beta \xi \right)+D_{3}\sinh\left(\kappa^{*}\beta \xi \right)+D_{4}\cosh\left(\kappa^{*}\beta \xi \right),\;1-\xi_{2}\leq \xi \leq 1, \end{array}\right. \end{array}$$
where $A_{i}$, $B_{i}$, $C_{i}$ and $D_{i}$ (i= 1–4) are the sixteen unknown constants to be determined by the sixteen boundary conditions. The dimensionless quantities of β, κ, $\kappa_{2}$ and $\kappa^{*}$ are defined as the following:(9)$$\beta =\sqrt{\omega },\;\kappa \;=\;\sqrt[{4}]{\frac{\alpha }{\gamma }},\;\kappa_{2}\;=\;\sqrt[{4}]{\frac{1+\Delta_{1}}{1+\Delta_{2}}},\;\kappa^{*}\;=\;\sqrt[{4}]{\frac{\alpha^{*}}{\gamma^{*}}}.$$

The sixteen boundary conditions of Equation (Equation 4) now become the dimensionless ones as follows: (10)$$\begin{array}{c} U\left(0\right)=0,\;\frac{\partial U}{\partial \xi }\left(0\right)=0,\;U^{*}\left(1\right)=0,\;\frac{\partial U^{*}}{\partial \xi }\left(1\right)=0,\\ U\left(\xi_{1}\right)=(U_{1}\xi_{1}),\;\frac{\partial U}{\partial \xi }\left(\xi_{1}\right)=\frac{\partial U_{1}}{\partial \xi }\left(\xi_{1}\right),\;U\left(\xi_{1}\right)=U_{2}\left(\xi_{1}\right),\;\frac{\partial U}{\partial \xi }\left(\xi_{1}\right)=\frac{\partial U_{2}}{\partial \xi }\left(\xi_{1}\right),\\ \gamma \frac{\partial^{2}U}{\partial \xi^{2}}\left(\xi_{1}\right)=\frac{\partial^{2}U_{1}}{\partial \xi^{2}}\left(\xi_{1}\right)+\left(1+\Delta_{2}\right)\frac{\partial^{2}U_{2}}{\partial \xi^{2}}\left(\xi_{1}\right),\;\gamma \frac{\partial^{3}U}{\partial \xi^{3}}\left(\xi_{1}\right)=\frac{\partial^{3}U_{1}}{\partial \xi^{3}}\left(\xi_{1}\right)+\left(1+\Delta_{2}\right)\frac{\partial^{3}U_{2}}{\partial \xi^{3}}\left(\xi_{1}\right),\\ U_{1}\left(1-\xi_{2}\right)=U^{*}\left(1-\xi_{2}\right),\;\frac{\partial U_{1}}{\partial \xi }\left(1-\xi_{2}\right)=\frac{\partial U^{*}}{\partial \xi }\left(1-\xi_{2}\right),\\ U_{2}\left(1-\xi_{2}\right)=U^{*}\left(1-\xi_{2}\right),\;\frac{\partial U_{2}}{\partial \xi }\left(1-\xi_{2}\right)=\frac{\partial U^{*}}{\partial \xi }\left(1-\xi_{2}\right),\\ \frac{\partial^{2}U_{1}}{\partial \xi^{2}}\left(1-\xi_{2}\right)+\left(1+\Delta_{2}\right)\frac{\partial^{2}U_{2}}{\partial \xi^{2}}\left(1-\xi_{2}\right)=\gamma^{*}\frac{\partial^{2}U^{*}}{\partial \xi^{2}}\left(1-\xi_{2}\right),\\ \frac{\partial^{3}U_{1}}{\partial \xi^{3}}\left(1-\xi_{2}\right)+\left(1+\Delta_{2}\right)\frac{\partial^{3}U_{2}}{\partial \xi^{3}}\left(1-\xi_{2}\right)=\gamma^{*}\frac{\partial^{3}U^{*}}{\partial \xi^{3}}\left(1-\xi_{2}\right). \end{array}$$

By substituting the solution forms of Equation (Equation 8) into the boundary conditions of (Equation 10), an eigenvalue problem is formulated by setting the determinant of a 16×16 matrix zero, which is a nonlinear transcendental equation to be solved. Significant computation effort can be saved by utilizing the clamped boundary conditions of the two overhangs. By substituting Equation (Equation 8) into the first four boundary conditions of Equation (Equation 10), the following solution forms are obtained after some manipulations
(11)$$\begin{array}{c} \left\{\begin{array}{c} U\;=\;A_{1}\left[\sin\left(\kappa \beta \xi \right)-\sinh\left(\kappa \beta \xi \right)\right]+A_{2}\left[\cos\left(\kappa \beta \xi \right)-\cosh\left(\kappa \beta \xi \right)\right],\;0\leq \xi \leq \xi_{1}\\ U_{1}\;=\;B_{1}\sin\left(\beta \xi \right)+B_{2}\cos\left(\beta \xi \right)+B_{3}\sinh\left(\beta \xi \right)+B_{4}\cosh\left(\beta \xi \right),\;\xi_{1}\leq \xi \leq 1-\xi_{2}\\ U_{2}\;=\;C_{1}\sin\left(\kappa_{2}\beta \xi \right)+C_{2}\cos\left(\kappa_{2}\beta \xi \right)+C_{3}\sinh\left(\kappa_{2}\beta \xi \right)+C_{4}\cosh\left(\kappa_{2}\beta \xi \right),\;\xi_{1}\leq \xi \leq 1-\xi_{2}\\ U^{*}\;=\;D_{1}\left[\sin\left(\kappa^{*}\beta \xi \right)+d_{31}\sinh\left(\kappa^{*}\beta \xi \right)+d_{41}\cosh\left(\kappa^{*}\beta \xi \right)\right]\\ \;+D_{2}\left[\cos\left(\kappa^{*}\beta \xi \right)+d_{32}\sinh\left(\kappa^{*}\beta \xi \right)+d_{42}\cosh\left(\kappa^{*}\beta \xi \right)\right],\;1-\xi_{2}\leq \xi \leq 1. \end{array}\right. \end{array}$$

Here, $d_{31}$, $d_{32}$, $d_{41}$ and $d_{42}$ are the four constants determined by the clamped boundary conditions of the two overhangs at ξ=0 and 1, which are given in Appendix A. By substituting Equation (Equation 11) into the twelve boundary conditions of Equation (Equation 10) at the connecting points of $\xi_{1}$ and $\xi_{2}$, the following equation is obtained:(12)$$K\tilde{V}=\tilde{Z}.$$

Here, K=K(β) is a 12×12 matrix and its elements are given in Appendix A. $\tilde{V}$ is a vector defined as ${\tilde{V}}^{T}$$=(A_{1},A_{1},B_{1},B_{2},B_{3},B_{4},C_{1},C_{2},C_{3},C_{4},D_{1},D_{2})$. $\tilde{Z}$ is the zero vector of ${\tilde{Z}}^{T}$=(0,0,0,0,0,0,0,0,0,0,0,0). By setting the determinant of K to zero, i.e., det(K) = 0, the eigenvalues (βs) are found numerically one by one from the lower order to the higher one by the Newton–Raphson method [51]. For most structures, their two adjacent eigenvalues are well-separated. In contrast, the two adjacent eigenvalues of an overhanging structure are very close to each other at the veering loci. Therefore, extreme caution during the computation should be taken when solving this nonlinear transcendental equation. Otherwise, wrongful results can be easily generated during the eigenfrequency computation. After an eigenvalue is found and substituted into the above equation, the constants of $A_{1}$, $A_{2}$, $A_{3}$, $A_{4}$, $B_{1}$, $B_{2}$, $B_{3}$, $B_{4}$, $C_{1}$, $C_{2}$, $C_{3}$, $C_{4}$ and $D_{1}$ can be found by setting $D_{2}=1$. Once those constants are found and substituted into Equation (Equation 11), the mode shape is also obtained. As given in Appendix A, K contains both diagonal and off-diagonal terms. Mathematically, those off-diagonal terms contain the coupling information [11,16,52]. Here, a noteworthy point is that though the eigenfrequency is numerically computed by setting det(K) = 0, there is no numerical discretization procedure involved.

## 3. Results and Discussion

In the above derivations for a general case, the material properties of the two overhangs and beams are different. For simplicity, we assume that they are made of a same material, i.e., $E=E^{*}=E_{1}=E_{2}$ and $\rho =\rho^{*}=\rho_{1}=\rho_{2}$, which is a common scenario of many micro/nano-resonators [34,35,36].

In Figure 2, the case of $b=b^{*}=6b_{1}=6b_{2}$, $h=h^{*}=h_{1}=h_{2}$ and $\Delta_{1}=\Delta_{2}=0$ is firstly examined. In this study, the two overhangs are with the same geometry except the one in Figure 9. The above parameters also indicate that the two beams are identical. In Figure 2, the variations of the first six $\beta_{n}$s (n=1–6) as functions of the overhang length ($\xi_{1}$) are presented. Here, $\beta_{n}=\sqrt{\omega_{n}}$ is the square root of the *n*th dimensionless eigenfrequency ($\omega_{n}$). In the vibration study, it is more common to use this β parameter rather than ω [50]. For the comparison reason, the first six eigenfrequency square roots of a uniform beam ($f_{i}$s) are also marked in Figure 2 and their values are given in Table 1. For the convenience of statement, instead of the eigenfrequency square root, we call $\beta_{n}$ and $f_{i}$ eigenfrequency hereafter. In Figure 2, a distinct feature arises: $\beta_{2i-1}$ and $\beta_{2i}$ (*i* = 1, 2 and 3) appear as a pair and then separate. When $\xi_{1}=\xi_{2}=0$, there is no overhang and the whole structure is two separate clamped–clamped beams without any coupling. Therefore, $\beta_{1}=\beta_{2}=f_{1}$, $\beta_{3}=\beta_{4}=f_{2}$ and $\beta_{5}=\beta_{4}=f_{3}$ are the first three eigenfrequencies of a uniform beam. As the overhang length increases, $\beta_{2i-1}$ and $\beta_{2i}$ begin to separate. When $\xi_{1}=\xi_{2}=0.5$, the two separate beams merge into one (uniform) beam. For a rectangular beam, its eigenfrequency $f_{i}\propto \sqrt{EI/(mL^{4})}\propto h/L^{2}$ [50], i.e., the beam width has no impact on its eigenfrequency, which is also the mathematical reason in the two cases of $\xi_{1}=\xi_{2}=0$ and $\xi_{1}=\xi_{2}=0.5$, $\beta_{i}$s match $f_{i}$s. These two cases are also the benchmark tests to see if the eigenfrequency computation is correctly carried out. When the overhang length ($\xi_{1}$) approaches 0.5, the beams are about to become one and the coupling is about to vanish, $\beta_{2i}$ approaches $f_{2i}$ and $\beta_{2i-1}$ approaches $f_{2i-1}$. In summary, $\beta_{2i-1}$ and $\beta_{2i}$ start with $f_{i}$ and they separate; finally, $\beta_{2i}$ ends up with $f_{2i}$, and $\beta_{2i-1}$ with $f_{2i-1}$. Physically, $\beta_{2i}$s are the newly emerging eigenfrequencies due to the coupling of the two beams by the overhangs. Similarly, new eigenfrequencies also arise in the acoustic and optical bands of a crystal due to the introduction of an impurity [1,2]. In optomechanics, several important applications are associated with this phenomenon of newly emerging eigenfrequency/mode shape and their separations, which corresponds to the optomechanically induced transparency [14].

The veering characteristics can be summarized from the results in Figure 2: higher eigenvalues experience more veering loci. For example, there is no veering locus for $\beta_{1}$; only one for $\beta_{2}$ marked as No. 1; two for $\beta_{3}$ marked as Nos. 1 and 2; three for $\beta_{4}$ marked as Nos. 2–4; and four for $\beta_{5}$ marked as Nos. 3–6. Because only up to six eigenfrequencies are plotted, there are only two $\beta_{6}$ veering loci marked as 5 and 6 shown in Figure 2. Actually there are three more $\beta_{6}$ veering loci as indicated by the arrows, which are formed by the $\beta_{6}/\beta_{7}$ veering. Physically, the variation of the overhang length ($\xi_{1}$) changes the mass and stiffness distribution of the system and its impact on different mode is different [19], which is clearly seen in the $\beta_{i}$ wavy variations in Figure 2. As the two overhangs here are identical, there is only one control parameter, i.e., $\xi_{1}$. The eigenfrequency variation as the function of $\xi_{1}$ is a curve and the corresponding veering is thus called curve veering [42,44]. When the number of the control parameter is two and the eigenvalue variation is a surface, the corresponding veering is thus called surface veering [12]. Six eigenfrequency veering loci are identified and marked with Arabic numbers in Figure 2. Locus No. 1 is the $\beta_{2}/\beta_{3}$ veering; No. 2 is the $\beta_{2}/\beta_{3}$ veering; Nos. 3 and 4 are the $\beta_{4}/\beta_{5}$ veering; and Nos. 4 and 5 are the $\beta_{5}/\beta_{6}$ veering. The smallest gap between two adjacent veering eigenvalues is called veering neck [53] or veering width [54]. Clearly, the veering necks of veering loci Nos. 1 and 3 are much smaller than the other three. Figure 3 presents a closer look at these two veering loci. Figure 3a plots the $\beta_{2}/\beta_{3}$ veering around $\xi_{1}=0.235$, at which the veering width $\beta_{3}-\beta_{2}\approx 0.06$ is achieved. Similarly, Figure 3b plots the $\beta_{4}/\beta_{5}$ veering around $\xi_{1}=0.165$, at which the veering width $\beta_{5}-\beta_{4}\approx 0.13$ is achieved. Figure 3 shows a distinctive feature of veering: The two adjacent eigenvalues rapidly converge and then abruptly diverge; however, they do not cross each other. Therefore, instead of veering, a more straightforward name of “avoided crossing” is used by physicists [14]. It is worth mentioning that eigenvalue crossing rather than veering can occur for an overhanging structure [49].

An important issue here is how to extract the effective strength of coupling from the the veering plots. In the modeling viewpoint, the coupling effect is incorporated in those six parameters as defined in Equation (Equation 7) as well as the two parameters of the overhang locations ($\xi_{1}$ and $\xi_{2}$). These parameters appear in both the governing equation of Equation (Equation 6) and boundary conditions of Equation (Equation 10), and, therefore, it is extremely difficult if not impossible to derive an analytical expression for the effective strength of coupling for this simple overhanging structure. A similar problem is also encountered in the model of internally coupled ears (ICE) [55,56], in which the two eardrums are modeled as two membranes connected by a cavity. In the ICE model, the coupling effect is determined by several parameters such as the tension, damping, radius, thickness and density of the membranes, and the air density, pressure and length of the cavity [55]. The vibration of the membrane poses a movable boundary problem for the propagation of acoustic wave inside the cavity [55,56], which further complicates the problem and thus the evaluation on the coupling effect. A major difficulty in the development of the overhanging resonator sensor is to control/characterize the initial disorders [40], which influence the coupling strength and result from the manufacturing error. However, those parameters in essence change the eigenfrequencies of a coupled system and the effective strength of coupling stiffness, $k_{c}$, can be effectively evaluated by the following equation [57]
(13)$$\frac{k_{c}}{k}=\frac{\omega_{2}^{2}-\omega_{1}^{2}}{2\omega_{1}^{2}},$$
where $\omega_{2}$ and $\omega_{1}$ are the two (measured) adjacent eigenfrequencies and $\omega_{2}>\omega_{1}$. *k* is the (oscillator) stiffness and in our case *k* is the cantilever effective stiffness [57], which is dependent on its mode(s) excited [52]. According to Equation (Equation 13), at the loci of the veering neck/width [53,54], $k_{c}$ reaches its minimum and thus indicates the weakest physical coupling. However, this minimum $k_{c}$, i.e., the weakest physical coupling, signifies the strongest mode coupling [15,17]. Generally speaking, large $k_{c}$ leads to the synchronization and small $k_{c}$ leads to the (mode) localization [58].

Figure 4 examines the variations of the mode shapes of $\beta_{2}$ and $\beta_{3}$ before and after the veering locus No. 1 around $\xi_{1}=0.235$. Figure 4a,b plots the $\beta_{2}$ mode shape variations at $\xi_{1}=0.21$ and $\xi_{1}=0.25$, respectively. Because two beams can vibrate differently, there are two different deflections, which are normalized by being divided by the maximum displacement of Beam 1. At $\xi_{1}=0.21$, the peak of Beam 1 (approximately) corresponds to the valley of Beam 2, which is called that the two beams are out-of-phase [38]. When $\xi_{1}=0.25$, a significant change occurs: Now, the peaks/valleys of Beam 1 (approximately) matches those of Beam 2, which is called that the two beams are in-phase [38]. Similarly, Figure 4c,d plots the $\beta_{3}$ mode shape variations at $\xi_{1}=0.21$ and $\xi_{1}=0.25$, respectively. Compared with those in Figure 4a,b, things are reversed: the $\beta_{3}$ mode shape experiences the transition of the in-phase mode to the out-of-phase mode. When Figure 4a,c is examined together, physically, it means that, for a given overhanging structure with $\xi_{1}=0.21$, the lower $\beta_{2}$ mode is out-of-phase and the higher $\beta_{3}$ mode is in-phase. In the other configuration of overhanging structure with $\xi_{1}=0.25$, as shown in Figure 4b,d, things are reversed again: the lower $\beta_{2}$ mode is in-phase and the higher $\beta_{3}$ mode is out-of-phase. This in-phase and out-of-phase pattern shift in the mode shape is often accompanied by significant amplitude variations [40,41].

Figure 5 examines the variations of the mode shapes of $\beta_{4}$ and $\beta_{5}$ before and after the veering locus No. 3 around $\xi_{1}=0.165$. Similar to the $\beta_{2}$ mode shape variation before and after a veering locus in Figure 4, the $\beta_{4}$ mode shape in Figure 5 experiences the transition of the out-of-phase mode to the in-phase mode. Meanwhile, the $\beta_{5}$ mode shape in Figure 5 experiences the transition of the in-phase mode to the out-of-phase mode, which is similar to the $\beta_{3}$ mode in Figure 4. In summary, before and after a veering locus, not only the two veering eigenvalues but also their mode shapes experience rapid changes. Because a small change of $\xi_{1}$ results in large changes of eigenvalue and mode shape, eigenvalue veering is a catastrophic type phenomenon [11]. Furthermore, the mode shape change from the in-phase mode to out-of-phase mode or vice versa is a qualitative one, which is expected to be of important use in the mass resonator sensors based on the mode shape variation [35,54].

Instead of just focusing the mode shape variations in a narrow area before and after a veering locus, Figure 6 and Figure 7 provide an overall and more comprehensive study on the mode shape variations. In Figure 6 and Figure 7, each row is the mode shapes of a $\beta_{i}$ with different $\xi_{1}$s of 0.1, 0.2, 0.3, 0.4 and 0.5. The values of $\beta_{i}$s are given in Table 1. For each row of a $\beta_{i}$ mode shape, the mode shapes with different $\xi_{1}$s are plotted in the same scale for the comparison reason. Figure 6 presents the mode shapes of the first three $\beta_{i}$s and Figure 7 presents those of the last three. Two patterns can be summarized from the results in these two figures. The first is that the odd mode of $\beta_{2i-1}$ except $\beta_{1}$ experiences the in-phase/out-of-phase mode transition around the first veering locus and the out-of-phase/in-phase transition around the second veering locus, and then the mode transitions of the in-phase/out-of-phase and out-of-phase/in-phase alternate. In contrast, the even mode of $\beta_{2i}$ reverses the transition patterns: $\beta_{2i}$ mode experiences the out-of-phase/in-phase mode transition around the first veering locus and the in-phase/out-of-phase transition around the second veering locus, and then the mode transitions of the out-of-phase/in-phase and in-phase/out-of-phase alternate. As discussed above, Figure 4a,b shows the $\beta_{2}$ out-of-phase/in-phase mode transition around its first and only veering locus marked as No. 1, which is around $\xi_{1}=0.235$. This $\beta_{2}$ out-of-phase/in-phase mode transition around $\xi_{1}=0.235$ is also captured in the $\beta_{2}$ mode at $\xi_{1}=0.2$ and that at $\xi_{1}=0.3$ as presented in the second row of Figure 6. However, as discussed in Figure 2, a higher eigenvalue/mode experiences more than one veering locus. For example, $\beta_{3}$ experiences two veering loci around $\xi_{1}=0.235$ marked as No. 1 and around $\xi_{1}=0.32$ marked as No. 2, as shown in Figure 2. Again, the $\beta_{3}$ mode in-phase/out-of-phase transition around the first veering locus of $\xi_{1}=0.235$ is captured both by Figure 4c,d and the $\beta_{3}$ modes at $\xi_{1}=0.2$ and 0.3, as presented in Figure 6. Around the second veering locus of $\xi_{1}=0.32$, the $\beta_{3}$ mode out-of-phase/in-phase transition is captured in the $\beta_{3}$ modes at $\xi_{1}=0.3$ and $\xi_{1}=0.4$ in Figure 6.

The second one is that the wave numbers of all the $\beta_{n}$ (*n* = 1–6) modes except the $\beta_{1}$ mode increase as $\xi_{1}$ varies from 0 to 0.5. The mechanism for this wave number increase can explained as follows: As shown in Figure 2, the $\beta_{2}$ value starts with $f_{1}$ when $\xi_{1}=0$ and ends up with $f_{2}$ when $\xi_{1}=0.5$. Correspondingly, the $\beta_{2}$ mode starts with the first mode of a uniform beam (the $f_{1}$ mode) when $\xi_{1}=0$ and ends up with the second mode of a uniform beam (the $f_{2}$ mode) when $\xi_{1}=0.5$. Similarly, the $\beta_{3}$ mode starts with the $f_{2}$ mode and ends as the $f_{3}$ mode, the $\beta_{4}$ mode starts with the $f_{2}$ mode and ends up with the $f_{4}$ mode, and so on. The exception case of the $\beta_{1}$ mode as shown in the first row of Figure 6 always keep the $f_{1}$ or $f_{1}$-like mode and the in-phase configuration. Again, it is explained by Figure 2: $\beta_{1}$ starts with $f_{1}$ and ends up with the same $f_{1}$.

In Figure 6 and Figure 7, there are two benchmark characteristics of the mode shapes worth pointing out. The first one is that, at $\xi_{1}=0.5$, the deflection difference of two beams vanishes. The reason is simple: the two beams with two overhangs merge into one uniform beam and, therefore, there is only one beam deflection. At ξ=0, the structure configuration is two separate and independent beams. Therefore, the mode shape can be in two scenarios: The first one is the in-phase configurations of $\xi_{1}=0.5$, as shown in Figure 6 and Figure 7, in which the two-beam deflections overlap. The second one is the out-of-phase configuration, in which the two-beam deflections are in the exactly opposite phases. The second characteristic is that except the value of $\xi_{1}=0.5$ where the two-beam deflections overlaps, the two-beam deflections are more or less different from other values of $\xi_{1}$. For example, in the first row of Figure 7, there is a huge difference between the two-beam deflections of the $\beta_{4}$ at $\xi_{1}=0.3$; in comparison, the difference is much smaller at $\xi_{1}=0.4$. This deflection difference of the two-beam in an eigenmode is called mode localization [35] or confinement of vibration [59]. Mode localization physically means that motion/energy is not equally distributed in each part of a system; the motion/energy concentration on one or some parts of a system can be harmful to cause an unexpected fatigue failure [9].

It is worth explaining briefly how the mode localization is induced in the overhanging structure. As mentioned above, the localized modes in locally resonant materials are realized in a structure consisting of two materials with huge difference [6]. In contrast, the two overhangs here are identical and two beams are also identical, a natural question arises: Why do the two beams vibrate differently? Many theories have been firmly established to show that the mode localization can also be caused by mistuning, i.e., irregularity of a periodic structure [9,11,16,59]. At the two connecting points of $\xi =\xi_{1}$ and $1-\xi_{2}$, there are discontinuities of the mass and bending stiffness, which leads to a sudden change of the governing equations as indicated by Equation (Equation 6). These discontinuities in essence are irregularities. When $\xi_{1}=0$ and 0.5, those irregularities are gone. When $\xi_{1}=0$, the two beams vibrate independently with the same shape; there is no coupling and thus no mode localization. When $\xi_{1}=0.5$, physically the two beams merge into one and mathematically the deflections of two beams overlap. Therefore, there is no mode localization, either. Overhang provides both the coupling mechanism and irregularity, which are responsible for the mode localization. The other worthy point is about the relation between the eigenfrequency veering and mode localization. Pierre presented the following statement: “Curve veering and strong mode localization are two manifestations of a single phenomenon.” [11]. In many structures, the curve veering and (strong) mode localization indeed occur together. In this overhanging structure, the simultaneous occurrence of eigenfrequency veering and mode location is also seen. For example, in the first row of Figure 7, there is a strong mode localization of the $\beta_{4}$ mode at $\xi_{1}=0.3$, which is near the veering locus No. 2 around $\xi_{2}=0.32$. A strong mode localization of the $\beta_{1}$ mode at $\xi_{1}=0.1$ is also seen. However, there is no eigenfrequency veering for $\beta_{1}$ at all, as shown in Figure 2. In contrast to Pierre’s viewpoint, Natsiavas [10] argued that the eigenfrequency veering and mode localization are not a same thing. In his analysis of a two-degree-freedom linear oscillator, Natsiavas [10] found that the mode localization occurs for the majority of the combinations of the system parameters and in contrast, the eigenfrequency veering only occurs within a very narrow range of the system parameters. In conjunction with the eigenfrequency veering loci in Figure 2 and mode shape variations in Figure 6 and Figure 7, we agree with Natsiavas on that in this particular overhanging structure, the eigenfrequency veering and mode localization are two different things; (strong) mode localization can occur without an eigenfrequency veering.

Figure 8 examines the impact of the overhang thickness on the eigenfrequency veering. Compared with the dimensions of the structure in Figure 2, all the geometric parameters are kept the same except that the overhang thickness is doubled as $h=h^{*}=2h_{1}=2h_{2}$. A distinctive difference between Figure 2 and Figure 8 is that $\beta_{i}$ now ends up with $\beta_{i}=\sqrt{2}f_{i}$ at $\xi_{1}=0.5$, although $\beta_{2i-1}$ and $\beta_{2i}$ still start with $f_{i}$. Again, at $\xi_{1}=0.5$, the overhanging structure becomes a uniform beam and its corresponding frequency is $\kappa \beta_{i}=f_{i}$. As defined in Equation (Equation 9), $\kappa =\sqrt[{4}]{\alpha /\gamma }=\sqrt[{4}]{h_{1}^{2}/h^{2}}=\sqrt{1/2}$. Therefore, $\beta_{i}=f_{i}/\kappa =\sqrt{2}f_{i}$. Another distinctive difference is that the number of the eigenfrequency veering is significantly reduced. In Figure 8, only three veering loci are identified. Figure 9 examines the asymmetry of the two overhang lengths on the eigenfrequency veering. Compared with the dimensions in Figure 8, all the geometric parameters are the same except that $\xi_{1}=1.5\xi_{2}$ is set in Figure 9. In all other figures, $\xi_{1}=\xi_{2}$ is set, which is a symmetric configuration. Now, with the asymmetric configuration, more veering loci arise in Figure 9. In conjunction with Figure 2, Figure 8 and Figure 9, it is clear that the geometry of the overhangs play the paramount role in the eigenfrequency veering, which determines the number of veering, veering loci and neck.

Figure 10 and Figure 11 study the influences of $\Delta_{1}$ and $\Delta_{2}$, respectively. In Figure 10 and Figure 11, $h=h^{*}=2h_{1}=2h_{2}$ and $\xi_{1}=0.265$ are set. In Figure 10, $\Delta_{2}=0$ is fixed and $\Delta_{1}$ varies from −0.2 to 0.2. Physically, $\Delta_{1}$ defined in Equation (Equation 7) is the difference of the mass per unit length between the two beams. Larger $\Delta_{1}$ means more mass of the system. Therefore, all $\beta_{i}$s in Figure 10 decrease monotonically with the increase of $\Delta_{1}$. In Figure 11, $\Delta_{1}=0$ is fixed and $\Delta_{2}$ varies from −0.2 to 0.2. $\Delta_{2}$ is also defined in Equation (Equation 7) and physically it is the difference of the bending stiffness between the two beams. Larger $\Delta_{2}$ means larger system stiffness. Therefore, all $\beta_{i}$s in Figure 11 increase monotonically with the increase of $\Delta_{2}$. From the results in Figure 10 and Figure 11, it is concluded that $\Delta_{1}$ and $\Delta_{2}$ can have significant impact on some eigenfrequencies of $\beta_{i}$s. The dramatic decrease and increase of $\beta_{5}$ in Figure 10 and Figure 11 are seen. However, there is no veering behavior of abrupt converging and then diverging. The variations of $\Delta_{1}$ and $\Delta_{2}$ only change the gap distance of two (adjacent) veering eigenfrequencies, i.e., $\beta_{i+1}-\beta_{i}$, and thus their veering neck. For example, the gap distance between $\beta_{2}$ and $\beta_{3}$ in Figure 10 decreases with the $\Delta_{1}$ increase. In contrast, the gap distance between $\beta_{2}$ and $\beta_{3}$ increases with the $\Delta_{2}$ increase in Figure 11. The setting of $\xi_{1}=0.265$ in Figure 10 and Figure 11 is close to the veering locus of $\xi_{1}=0.28$ marked as No. 1 in Figure 8, which is the $\beta_{2}/\beta_{3}$ veering. It is also noticed that, as $\Delta_{1}$ or $\Delta_{2}$ varies, the gap distance between $\beta_{3}$ and $\beta_{2}$ is always kept the smallest among all the gap distances.

## 4. Conclusions

The governing equations of the doubly clamped overhang-two beams-overhang structure are derived and the corresponding eigenvalue problem is formulated, whose computation requires no numerical discretization. The variations of the eigenfrequency and mode shape are studied. The phenomena of the eigenfrequency curve veering, mode splitting and mode localization are shown. Besides coupling two beams, the presence of overhang also changes the spatial distribution of mass and stiffness of the system, which has different impact on the eigenfrequency of different mode. The eigenfrequencies of different modes vary differently with the varying length of the overhang, which leads to eigenfrequency curving veerings at some given overhang lengths. Before and after a veering locus, both the eigenfrequencies and mode shapes of the two adjacent veering modes experience catastrophic changes. The out-of-phase/in-phase transition and in-phase/out-of-phase transition of the mode shapes of two veering modes before and after a veering locus are shown. In many structures, the phenomena of eigenfrequency curve veering and mode localization occur simultaneously. In our structure, they can occur simultaneously or the mode localization can occur alone without any eigenfrequency curve veering. The geometric parameters of overhang are the paramount parameters of determining the eigenfrequency veering; the mass and stiffness differences between the two beams are the tuning parameters, which vary the veering neck.

## Figures and Tables

**Figure 1 micromachines-12-00324-f001:**
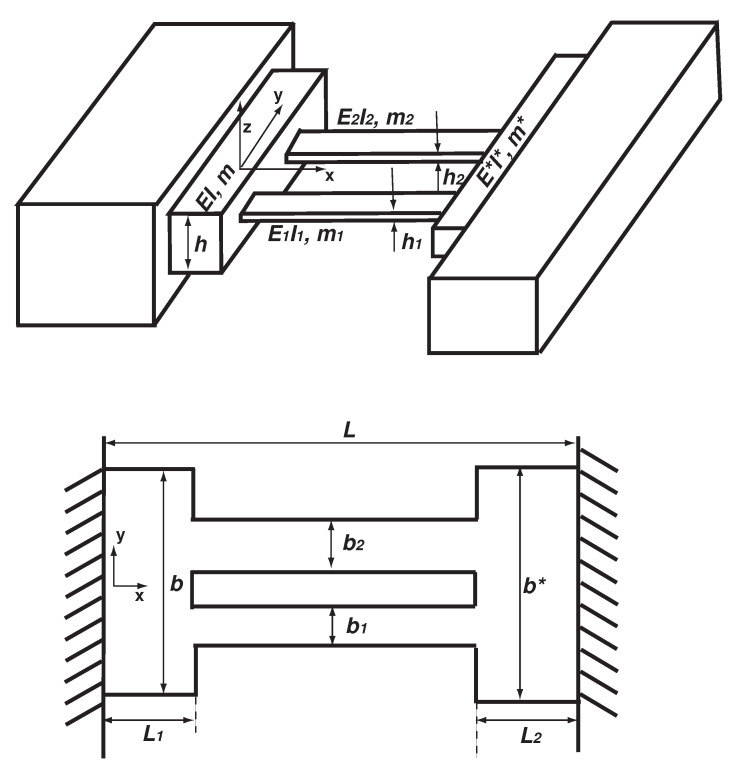
The schematic diagram of the doubly clamped overhang-two beams-overhang structure and its dimensions. Here, *x* and *y* are the length and width directions, respectively. The structure vibrates transversely in the *z*-direction.

**Figure 2 micromachines-12-00324-f002:**
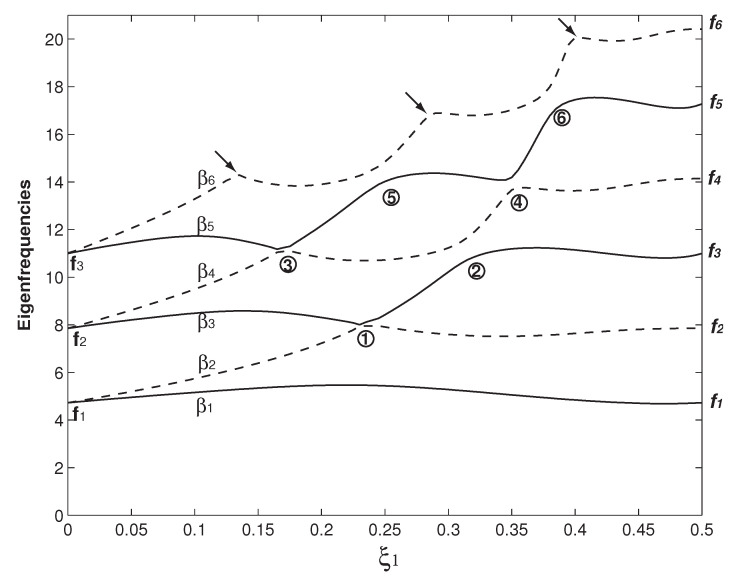
The first six eigenfrequencies ($\beta_{n}$, n=1 to 6) as the functions of the overhang length ($\xi_{1}$) for two identical beams, i.e., $m_{1}=m_{2}$ and $E_{1}I_{1}=E_{2}I_{2}$. The dimensions of the two overhangs are also set the same as $L_{1}=L_{2}$, $b=b^{*}=6b_{1}=6b_{2}$ and $h=h^{*}=h_{1}=h_{2}$. The corresponding dimensionless parameters are α=γ=6, $\alpha^{*}=\gamma^{*}=6$ and $\Delta_{1}=\Delta_{2}=0$, respectively. Here, $f_{i}$s (i=1–6) are the first six dimensionless eigenfrequencies of a uniform clamped–clamped beam. The veering loci are marked by Arabic numbers.

**Figure 3 micromachines-12-00324-f003:**
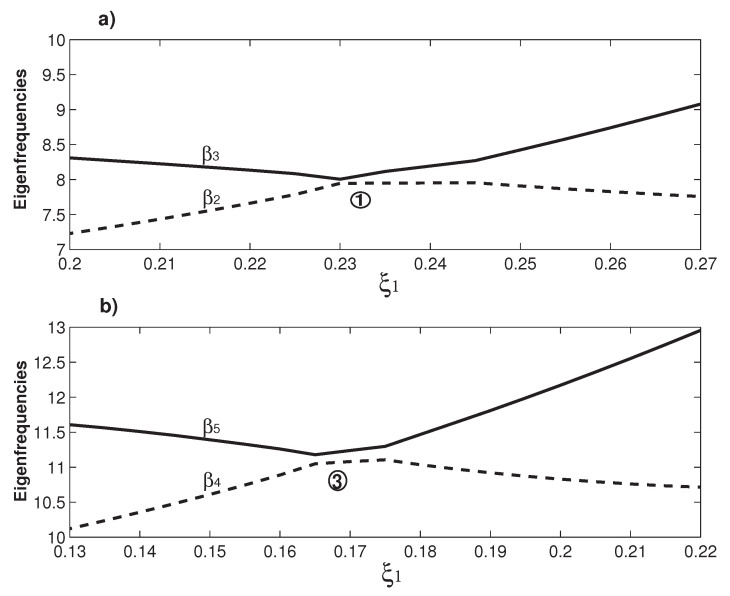
(**a**) A closer look at the veering of $\beta_{2}/\beta_{3}$ around $\xi_{1}=0.23$, which is marked as the No. 1 veering locus in Figure 2; and (**b**) a closer look at the veering of $\beta_{4}/\beta_{5}$ around $\xi_{1}=0.17$, which is marked as the No. 2 veering locus in Figure 2.

**Figure 4 micromachines-12-00324-f004:**
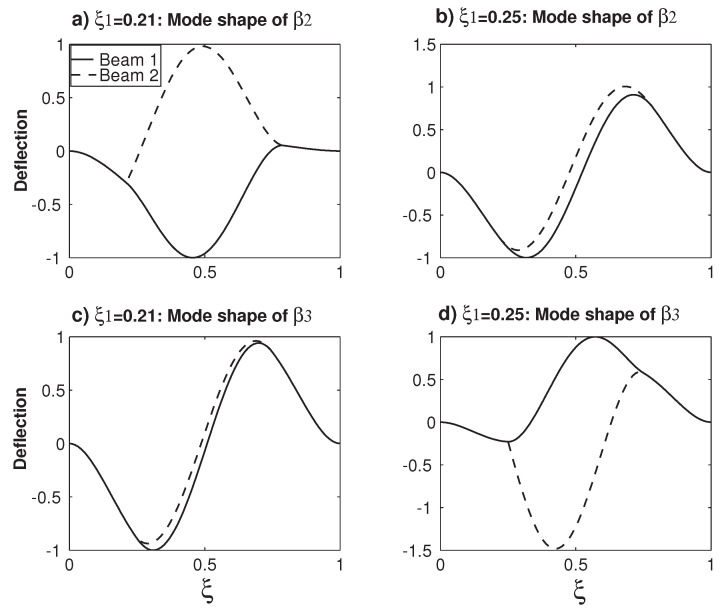
The changes of the mode shapes before and after a veering locus. (**a**) The $\beta_{2}$ mode shape at $\xi_{1}=0.21$ and (**b**) the $\beta_{2}$ mode shape at $\xi_{1}=0.25$. (**c**) The $\beta_{3}$ mode shape at $\xi_{1}=0.21$ and (**d**) the $\beta_{3}$ mode shape at $\xi_{1}=0.25$.

**Figure 5 micromachines-12-00324-f005:**
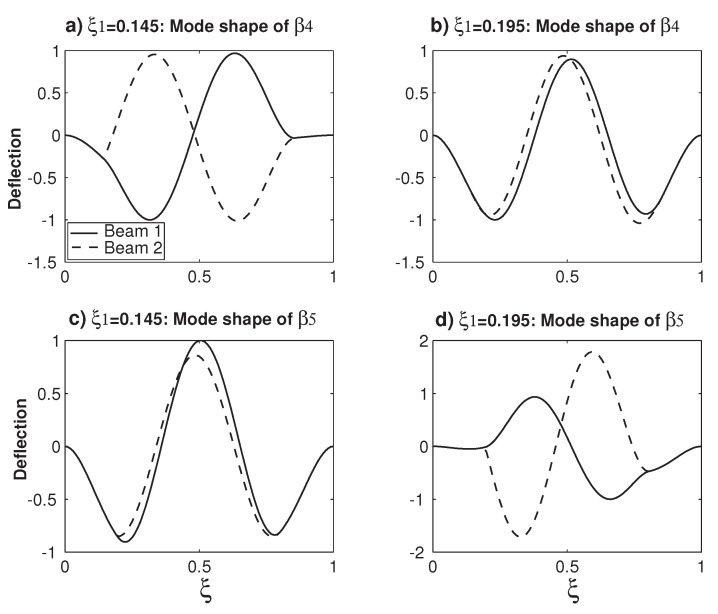
The changes of the mode shapes before and after a veering locus. (**a**) The $\beta_{4}$ mode shape at $\xi_{1}=0.145$ and (**b**) the $\beta_{4}$ mode shape at $\xi_{1}=0.195$. (**c**) The $\beta_{5}$ mode shape at $\xi_{1}=0.145$ and (**d**) the $\beta_{5}$ mode shape at $\xi_{1}=0.195$.

**Figure 6 micromachines-12-00324-f006:**
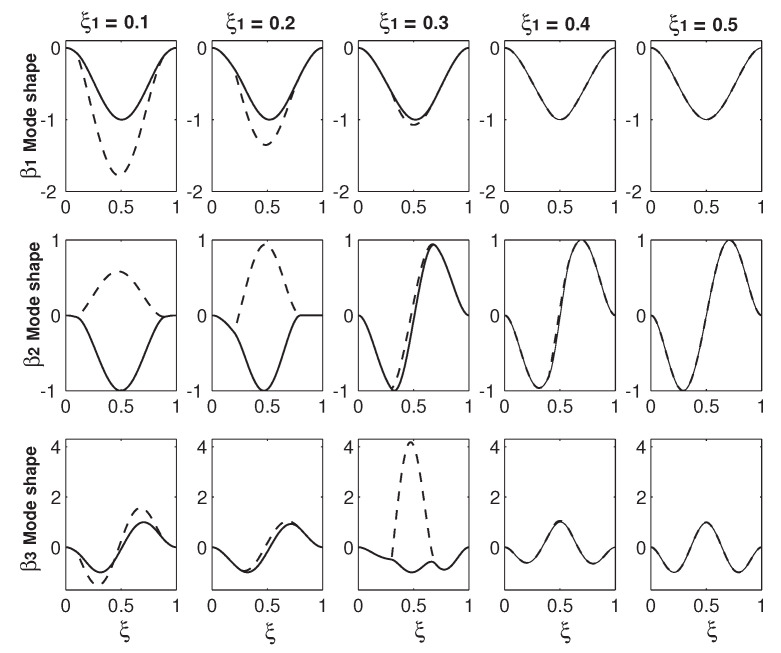
The mode shape variations of the first three eigenfrequencies ($\beta_{1}$, $\beta_{2}$ and $\beta_{3}$) as the functions of the overhang length $\xi_{1}$. The first, second and third rows correspond to the mode shapes of $\beta_{1}$, $\beta_{2}$ and $\beta_{3}$, respectively.

**Figure 7 micromachines-12-00324-f007:**
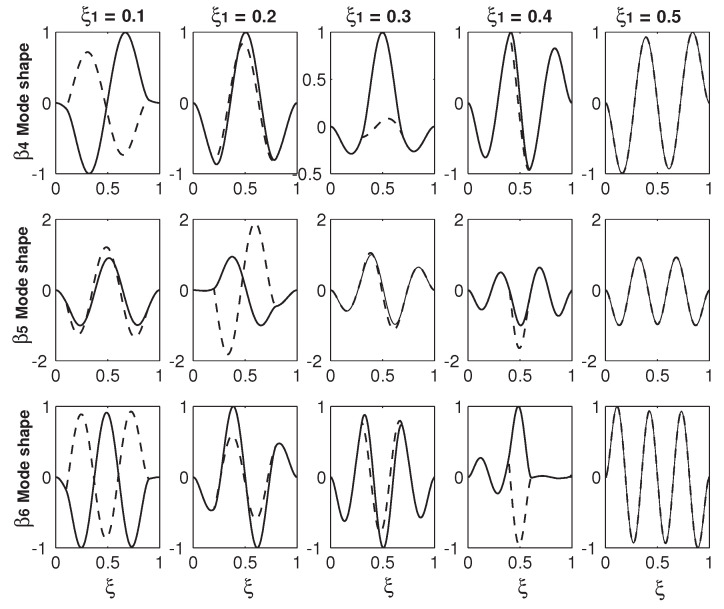
The mode shape variations of the last three eigenfrequencies ($\beta_{4}$, $\beta_{5}$ and $\beta_{6}$) as the functions of the overhang length $\xi_{1}$. The first, second and third rows correspond to the mode shapes of $\beta_{4}$, $\beta_{5}$ and $\beta_{6}$, respectively.

**Figure 8 micromachines-12-00324-f008:**
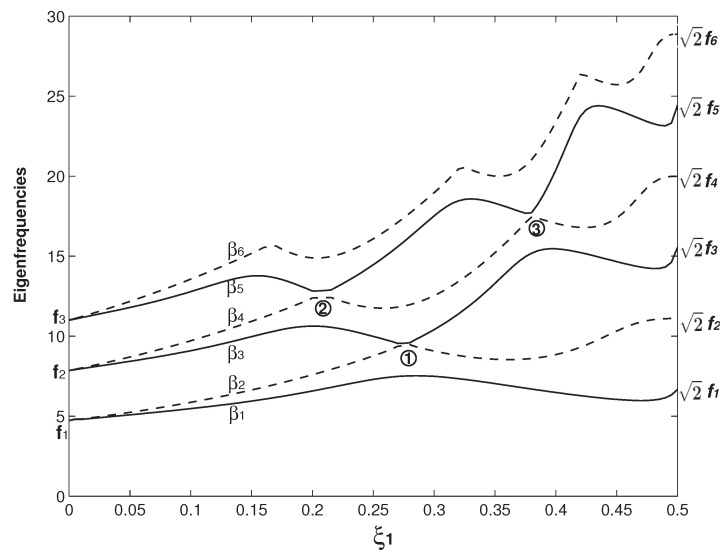
The first six eigenfrequencies ($\beta_{n}$, n=1–6) as the functions of the overhang length ($\xi_{1}$) for two identical beams, i.e., $m_{1}=m_{2}$ and $E_{1}I_{1}=E_{2}I_{2}$. The dimensions of the two overhangs are also set the same as $L_{1}=L_{2}$, $b=b^{*}=6b_{1}=6b_{2}$ and $h=h^{*}=2h_{1}=2h_{2}$. Compared with the dimensions of Figure 2, the overhang thickness is doubled. The corresponding dimensionless parameters are $\alpha =\alpha^{*}=12$, $\gamma =\gamma^{*}=48$ and $\Delta_{1}=\Delta_{2}=0$, respectively.

**Figure 9 micromachines-12-00324-f009:**
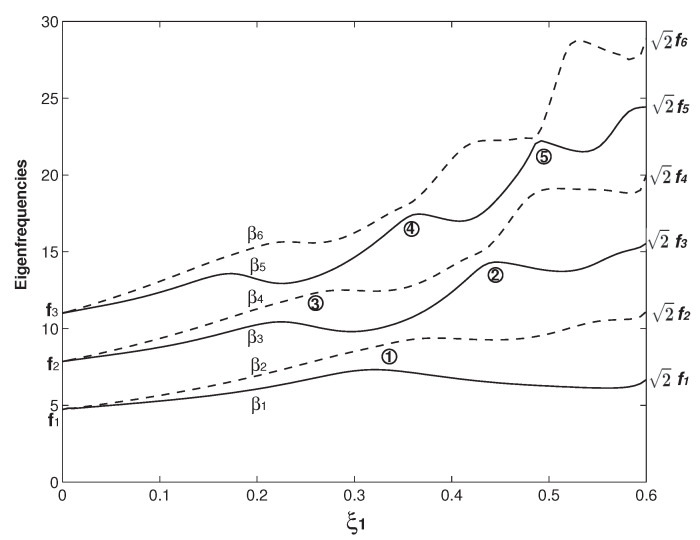
The impact of the overhang asymmetry on the eigenfrequencies. Compared with the dimensions in Figure 8, all the dimensions are kept the same except the overhang lengths. The asymmetric configuration of $\xi_{1}=1.5\xi_{2}$ is set.

**Figure 10 micromachines-12-00324-f010:**
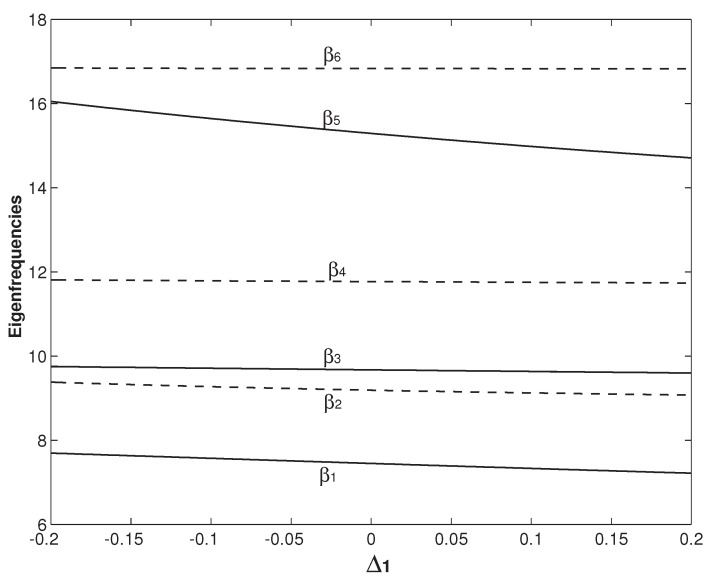
The first six eigenfrequencies as the functions of $\Delta_{1}$. The overhang length is fixed as $\xi_{1}=\xi_{2}=0.265$, which is around the veering locus No. 1 in Figure 8. Except $\Delta_{1}$, $\xi_{1}$ and $\xi_{2}$, all other parameters are the same as those in Figure 8.

**Figure 11 micromachines-12-00324-f011:**
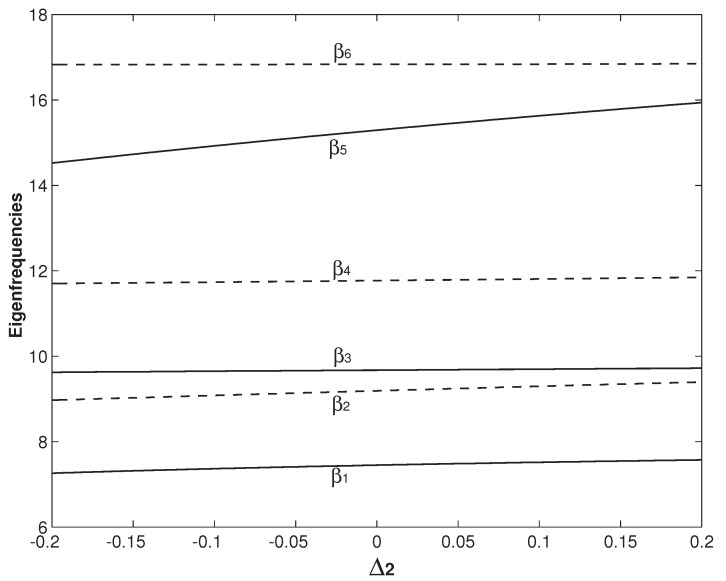
The first six eigenfrequencies as the functions of $\Delta_{2}$ with the overhang length being set as $\xi_{1}=\xi_{2}=0.265$.

**Table 1 micromachines-12-00324-t001:** The first six eigenfrequencies ($f_{i}$s) of a uniform beam [50] and first six eigenfrequencies ($\beta_{n}$s) of the overhanging structure in Figure 2 at $\xi_{1}=0.1$, 0.2, 0.3, 0.4 and 0.5.

Uniform	$f_{1}$	$f_{2}$	$f_{3}$	$f_{4}$	$f_{5}$	$f_{6}$
Beam	4.73	7.853	10.996	14.137	17.279	20.42
Overhanged	$\beta_{1}$	$\beta_{2}$	$\beta_{3}$	$\beta_{4}$	$\beta_{5}$	$\beta_{6}$
$\xi_{1}=0.1$	5.161	5.742	8.488	9.496	11.728	13.286
$\xi_{1}=0.2$	5.458	7.228	8.309	10.83	12.171	13.902
$\xi_{1}=0.3$	5.283	7.598	10.224	11.229	14.345	16.865
$\xi_{1}=0.4$	4.845	7.639	11.144	13.631	17.455	20.059
$\xi_{1}=0.5$	4.73	7.853	10.996	14.137	17.279	20.42

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
