# Peer review of "Mode Localization and Eigenfrequency Curve Veerings of Two Overhanged Beams"

_micromachines, 2021, doi:10.3390/mi12030324_

Round 1

Reviewer 1 Report

Mode Localization and Eigenfrequency Curve Veerings of Two Overhanged Beams

  1. Zhang, Y. Petrov, and Y.-P. Zhao

The authors present the analytical derivation of the governing equations of doubly clamped beams connected by an overhang. By solving these equations, they have studied the eigenfrequency and mode changes as a function of different parameter variations.

The manuscript is clear and well written. I think it could be of interest for the community. However, prior my final recommendation I have some comments that must be taken into consideration.

  1. In the introduction section the authors made a thorough review of the mode localization. However, in the acoustic section when they say “In acoustics, the localized modes are realized by the so-called locally resonant sonic materials, which consist of two different materials: Hard lead particle coated with soft silicone rubber [6]”. The periodic modulation of the elasticity modulus is a phononic crystal, an emerging field that deserves a brief explanation.
  2. Please, avoid the use of phrases like “However, most of structures with small linear vibrations do not have the harmonic property and special design is required to achieve such property [19]”. This sentence is very vague, all the structures, especially in nanomechanics, need a “special design”.
  3. On page 4, the authors start to review the optomechanics field. The authors use “damping or negative damping” when talking about “cooling or amplification” please, be accurate when choosing the words because it is not the same. In optomechanics the interplay in between the optical and mechanical energy allows the bi-directional energy exchange. It is not an external damping.
  4. “Quality factor is an essential factor determining the mass sensitivity in a mass resonator sensor applications”. Actually, Q is related with the energy dissipation, and, thus, the noise in the frequency fluctuations.
  5. “When materials of the mechanical end mirror are highly transparent to light, it results in poor optomechanical coupling”. Actually, the optomechanical coupling depends on the spatial overlapping of the optical and mechanical modes. Be careful with the used language.
  6. When revisiting the mechanical coupled sensors, the authors forgot to mention that it is possible not only to track the frequency, but also the amplitude of the different modes, which is directly related with the localization. Few examples: https://doi.org/10.1088/0960-1317/25/9/095017,https://doi.org/10.1021/nl902350b, https://doi.org/10.1016/j.sna.2011.12.032
  7. There is a typo in page 7, when defining I=b*h*^3/12, I guess it should be I*
  8. Typo in the y-axis label in Figures 2 and 8 it should be “Eigenfrequencies”.

Author Response

Reviewer 1

The authors present the analytical derivation of the governing equations of doubly clamped beams connected by an overhang. By solving these equations, they have studied the eigenfrequency and mode changes as a function of different parameter variations.The manuscript is clear and well written. I think it could be of interest for the community. However, prior my final recommendation I have some comments that must be taken into consideration.

 Response: We are honored by the general assessment and detailed comments by the reviewer, which is of great help for us to improve the presentation of the manuscript. Thank you.

  1. In the introduction section the authors made a thorough review of the mode localization. However, in the acoustic section when they say “In acoustics, the localized modes are realized by the so-called locally resonant sonic materials, which consist of two different materials: Hard lead particle coated with soft silicone rubber [6]”. The periodic modulation of the elasticity modulus is a phononic crystal, an emerging field that deserves a brief explanation.

Response: As suggested by the reviewer, we add the following words to further explain this locally resonant sonic materials. “The transmission coefficient of the lead-rubber composite structure has the sudden decreases at two different frequencies of an incidence wave, which are referred to as the dip frequencies [6]. At the lower dip frequency, the lead particle is like a mass in an oscillator, which moves as a whole like a rigid body; at the same time the rubber displacements except those around the interface are very small. At the higher dip frequency, things are reversed: The lead particle now almost has no motion and the most of vibration is in the rubber. The motions at these two dip frequencies are referred to as the
localized modes [6]. In these localized modes, most of vibrations are confined either in the rubber layer or as the oscillation of the lead particle[6]; almost no waves are transmitted into the lead particle, or say, the particle has no deformation. This localized mode provides another effective way of sound attenuation rather than absorption [6].”

  1. Please, avoid the use of phrases like “However, most of structures with small linear vibrations do not have the harmonic property and special design is required to achieve such property [19]”. This sentence is very vague, all the structures, especially in nanomechanics, need a “special design”.

Response: As suggested by the reviewer, we make the statement more specific as: “various methods, such as the step-like design [19], are used to make a cantilever beam harmonic.”

  1. On page 4, the authors start to review the optomechanics field. The authors use “damping or negative damping” when talking about “cooling or amplification” please, be accurate when choosing the words because it is not the same. In optomechanics the interplay in between the optical and mechanical energy allows the bi-directional energy exchange. It is not an external damping.

Response: I see the reviewer’s point.  Indeed, in this coupling system the energy transfer is two-way.When talking about the negative damping, we should specify which mode, as quoted from ref. [24] as follows:“Focussing on the mechanical mode alone, the growth of energy can be interpreted as “anti-damping” or amplification.” We revise the manuscript accordingly to specify the mode. Furthermore, negative damping is often used by the mechanical engineers and anti-damping is the term used by the physicists, which may have the minute difference and may also cause less confusion. We change all the negative damping terms as the anti-damping. The revised is paste for the convenience of the reviewers : “The coupling between the optical and mechanical modes can result in either the damping or anti-damping
force for either mode [24].”

  1. “Quality factor is an essential factor determining the mass sensitivity in a mass resonator sensor applications”. Actually, Q is related with the energy dissipation, and, thus, the noise in the frequency fluctuations.

Response: Indeed, Q indicates the energy dissipation. In the steady-state, Q influences two things: The amplitude and phase angle. In an approximate sense, Q is the amplification factor, which the ratio of the amplitude at the resonance as compared with that of static loading.  When considering the thermal fluctuation as the dominant noise source, Q is indeed a major factor determining the sensitivity and the minimum detectable mass in a mass resonator sensor. Besides the references cited in the manuscript, there are other references supporting this claim as seen in the following references: 1. E. Buks and B. Yurke, Mass detection with a nonlinear nanomechanical resonator, PHYSICAL REVIEW E 74, 046619 2006; 2. S. Schmid, L.G. Villanueva and M.L. Roukes, Fundamentals of Nanomechanical Resonators, Springer, 2016.

  1. “When materials of the mechanical end mirror are highly transparent to light, it results in poor optomechanical coupling”. Actually, the optomechanical coupling depends on the spatial overlapping of the optical and mechanical modes. Be careful with the used language.

Response: Indeed, the above statement is our paraphrase of reference [31]. We reread the reference and rephrase our statement, which should be more accurate to reflect the viewpoint by the authors of reference [31]. I think that the reviewer is correct on the coupling issue. But the authors of reference [31] still offer some straightforward but not that strict description on the coupling, which is beneficial for people like us. The revised texts are pasted here for the convenience of the reviewer: “When the materials of the mechanical end mirror are highly transparent to light, the radiation pressure exerted by photons can be weak, which is the means of the optomechanical coupling [31].”

  1. When revisiting the mechanical coupled sensors, the authors forgot to mention that it is possible not only to track the frequency, but also the amplitude of the different modes, which is directly related with the localization. Few examples: https://doi.org/10.1088/0960-1317/25/9/095017,https://doi.org/10.1021/nl902350b, https://doi.org/10.1016/j.sna.2011.12.032

Response: We thank the reviewer for providing these useful literatures and we cite two of them as refs [40] and [42] in the revised mansucript. Besides the amplitude variations, we also cite the two papers in other places.

  1. There is a typo in page 7, when defining I=b*h*^3/12, I guess it should be I*

Response: We are grateful to the reviewer for pointing out this typo and we fixed it.

  1. Typo in the y-axis label in Figures 2 and 8 it should be “Eigenfrequencies”.

Response: Again, we are grateful to the reviewer for pointing out these two typos and we fixed them.

Reviewer 2 Report

This paper is untitled Mode Localization and Eigenfrequency Curve Veerings of Two Overhanged Beams. In mathematics, and more particularly in linear algebra, we can remind that the concept of eigenvector is an algebraic notion applying to a linear map of a space in itself. It therefore corresponds to the study of the privileged axes, according to which the application behaves like a dilation, multiplying the vectors by the same constant. This dilation ratio is called eigenvalue, the vectors to which it applies are called eigenvectors, united in an eigen space. In this paper it is question of veerings of the Eigenfrequency and its effect on the errerror associated with the numerical discretization procedure. This paper present a method of computing the Eigenfrequency of two over hanged beams. The subject developed in this proposed article is very interesting in itself.

To start the actual review, I can say that the summary of the paper is well written and that it makes you want to read the article. He gives a good account of the contents of this paper. There is nothing to change.

About the introduction, I read it carefully, and I realize that the authors could have been a little more verbose about Lord Rayleigh. For this genius who received the Nobel Prize for Physics in 1904, it should be noted that all his publications and his writings are in the public domain. So I kindly invite authors to make the effort to cite his work. His works are readily available on the internet. Published by University Press (Cambridge), for example, it is possible to find them at the following link http://hdl.handle.net/1908/1406 and why not, go and find the corresponding references. Still, I really appreciate that the authors cited the main papers, and if not forgetting to cite Baron Rayleigh himself in the list of references, the rest is correct and sufficiently comprehensive. The bibliographical work carried out by the authors in this introduction is remarkable. It is really fun to read. It is very well documented.

Let us have a look at the model development. From the start, Figure 1 is appreciable with this 3D view and the top view, which allows you to represent what the different parameters correspond to which allow starting from the kinetic energy equation. Then it seems to me that the authors correctly applied Hamilton's principle. I manage to follow the reasoning and the theoretical calculations which govern this part 2. I do not see any errors or elements which would let me think that the reasoning would be inaccurate. This is why I validate the whole of this part devoted to Model Development. I also appreciate that the authors have indicated the matrix K in the appendix. It would have been too cumbersome to include it directly in the text of the paper.

Now here are some thoughts on Part 3 on Results and Discussion. The results are well illustrated through a series of several figures. The legends of the figures are relevant. Here again, I find that the paper can be read easily, with great fluidity. I also appreciate being able to see these values in table 1. Anyway, I want to stress that I like this part too. But I still have a small question that I ask myself. Looking at the results - for example, those presented in Table 1 - I find that the number of digits presented appeals to me. I want to ask the authors a question. Did the authors consider the validity of such precision in a real case? In other words, could the authors give some leads for an evaluation of the uncertainty associated with the values they give? The authors must be reassured, I do not ask them for a detailed calculation of the associated uncertainty but to give leads, and to say what, according to them, could influence the uncertainty associated with the given values?

I'll be faster on the conclusion. Here again, I find the authors to be precise and give a good conclusion, so I prefer to send them my congratulations on this conclusion.

I did not find any errors in the reference list. However, I invite the authors to make a final correction themselves.

I now come to the conclusion of this review. As others have seen, I have only two questions or recommendations. The first on actually getting to quote from a Lord Rayleigh paper. The second on a reflection on the potential uncertainty that would be associated with digital data. But I want to stress that I really enjoyed reading this paper. I find it fluid, well written and pleasant to read. My recommendation is that this paper has every legitimacy to be accepted for publication. Maybe I would put it accepted with minor revision, only because I think that the authors can still improve the quality of their paper a little bit. Once again, I give my congratulations and encouragement to the authors for their remarkable work.

Author Response

Reviewer 2

( )

( )

( )

( )

Comments and Suggestions for Authors

This paper is untitled Mode Localization and Eigenfrequency Curve Veerings of Two Overhanged Beams. In mathematics, and more particularly in linear algebra, we can remind that the concept of eigenvector is an algebraic notion applying to a linear map of a space in itself. It therefore corresponds to the study of the privileged axes, according to which the application behaves like a dilation, multiplying the vectors by the same constant. This dilation ratio is called eigenvalue, the vectors to which it applies are called eigenvectors, united in an eigen space. In this paper it is question of veerings of the Eigenfrequency and its effect on the error associated with the numerical discretization procedure. This paper present a method of computing the Eigenfrequency of two over hanged beams. The subject developed in this proposed article is very interesting in itself.

  1. To start the actual review, I can say that the summary of the paper is well written and that it makes you want to read the article. He gives a good account of the contents of this paper. There is nothing to change.

 Response: We are flattered by the reviewer’s comment and it is also a great honor to receive the comment like this. Thank you.

  1. About the introduction, I read it carefully, and I realize that the authors could have been a little more verbose about Lord Rayleigh. For this genius who received the Nobel Prize for Physics in 1904, it should be noted that all his publications and his writings are in the public domain. So I kindly invite authors to make the effort to cite his work. His works are readily available on the internet. Published by University Press (Cambridge), for example, it is possible to find them at the following link http://hdl.handle.net/1908/1406 and why not, go and find the corresponding references. Still, I really appreciate that the authors cited the main papers, and if not forgetting to cite Baron Rayleigh himself in the list of references, the rest is correct and sufficiently comprehensive. The bibliographical work carried out by the authors in this introduction is remarkable. It is really fun to read. It is very well documented.

Response: We are grateful to this information. I do not know that such collection of Lord Rayleigh’s work is available on Internet. What I had is just his classical book of Theory of Sound as cited as reference 3. Thanks for the information for the website and I downloaded it. With the given time of five days, it is impossible for us to read and grasp the research works of Lord Rayleigh. We have to stick to the status quo of only referencing one work of Lord Rayleigh as ref. [3] and we hope that the reviewer can forgive us on this issue.. 

  1. Let us have a look at the model development. From the start, Figure 1 is appreciable with this 3D view and the top view, which allows you to represent what the different parameters correspond to which allow starting from the kinetic energy equation. Then it seems to me that the authors correctly applied Hamilton's principle. I manage to follow the reasoning and the theoretical calculations which govern this part 2. I do not see any errors or elements which would let me think that the reasoning would be inaccurate. This is why I validate the whole of this part devoted to Model Development. I also appreciate that the authors have indicated the matrix K in the appendix. It would have been too cumbersome to include it directly in the text of the paper.

 Now here are some thoughts on Part 3 on Results and Discussion. The results are well illustrated through a series of several figures. The legends of the figures are relevant. Here again, I find that the paper can be read easily, with great fluidity. I also appreciate being able to see these values in table 1. Anyway, I want to stress that I like this part too. But I still have a small question that I ask myself. Looking at the results - for example, those presented in Table 1 - I find that the number of digits presented appeals to me. I want to ask the authors a question. Did the authors consider the validity of such precision in a real case? In other words, could the authors give some leads for an evaluation of the uncertainty associated with the values they give? The authors must be reassured, I do not ask them for a detailed calculation of the associated uncertainty but to give leads, and to say what, according to them, could influence the uncertainty associated with the given values?

Response: We understand the concern of the reviewer on this issue. Firstly, the model does not incorporate the damping, which more or less decreases the (squared) eigenfrequencies as presented in Table 1. Secondly, physically it is impossible to assume the beams and overhangs are identical in a practical application because there are always manufacturing errors. However, Table 1 together with Fig. 2 can at least offer a qualitative insight into the design of an overhanged structure. Thirdly and most importantly, Table 1 serves an implicit purpose of validating the results as presented in Fig. 2. The dimensionless (squared) eigenfrequencies of a uniform beam are the benchmark values of verifying our computation as those dimensionless values are available almost in every textbook. At \xi=0 (physically there is no overhang) and \xi=0.5(physically two beams merge into one and the eigenfrequencies are independent of the width), the corresponding eigenfrequencies are those of a uniform beam, which is verified in Fig. 2.

  1. I'll be faster on the conclusion. Here again, I find the authors to be precise and give a good conclusion, so I prefer to send them my congratulations on this conclusion. I did not find any errors in the reference list. However, I invite the authors to make a final correction themselves.

Response: We double checked the reference.

  1. I now come to the conclusion of this review. As others have seen, I have only two questions or recommendations. The first on actually getting to quote from a Lord Rayleigh paper. The second on a reflection on the potential uncertainty that would be associated with digital data. But I want to stress that I really enjoyed reading this paper. I find it fluid, well written and pleasant to read. My recommendation is that this paper has every legitimacy to be accepted for publication. Maybe I would put it accepted with minor revision, only because I think that the authors can still improve the quality of their paper a little bit. Once again, I give my congratulations and encouragement to the authors for their remarkable work.

Response: Again, we are extremely thankful and encouraged/honored by the comments of the reviewer. We are sorry that we did not address the first issue/question as given by reviewer. Again, we ask for the forgiveness and understanding of the reviewer on this issue/question.
